# SPIKE-RL: Video-LLMs meet Bayesian Surprise

**Sahithya Ravi** [1,2]  **Aditya Chinchure** [1,2]  **Raymond T. Ng** [1]  **Leonid Sigal** [1,2]  **Vered Shwartz** [1,2]

[1] The University of British Columbia,  [2] Vector Institute for AI
{sahiravi, aditya10, rng, lsigal, vshwartz}@cs.ubc.ca

## Abstract

Real-world videos often show routine activities punctuated by memorable, surprising events. However, most Video-LLMs process videos by sampling frames uniformly, likely missing critical moments that define a video's narrative. We introduce SPIKE, an inference-time framework that quantifies Bayesian Surprise as the belief update triggered by new visual evidence in the video stream, identifying moments where new visual evidence conflicts with prior beliefs. SPIKE effectively localizes surprise in videos, correlated with humans on positive (FunQA) and negative (Oops!) surprise benchmarks. SPIKE-RL further improves on SPIKE's ability to detect surprise, leveraging GRPO to refine its belief hypotheses based on a reward signal from the video caption. SPIKE and SPIKE-RL guide query-agnostic surprise-weighted frame sampling, which allocates more frames to interesting moments in the video. With this strategy, we achieve consistent performance gains on five downstream benchmarks. By enabling Video-LLMs to track beliefs and register surprise, our work paves the way for more robust models that can revise their understanding in response to new information.[1]

## 1 Introduction

Humans navigate the world not as passive observers, but as active predictors of the future who infer the hidden causes behind events and update their predictions (Millidge et al., 2022). This process, formalized within the Bayesian Theory of Mind (ToM) framework (Baker et al., 2017), suggests that our brain continuously builds and updates an internal model of the world, using discrepancies between expectation and reality, or *surprise*, as the primary signal for learning and attention. This allows us to efficiently process a constant stream of sensory data, focusing our cognitive resources on moments that are novel and informative, and ignoring redundant, expected information. For instance, in the Mr. Bean video shown in Figure 1, our cognitive focus is on the moment the man unexpectedly falls, because it deviates from the established routine.

However, current Video-LLMs are fundamentally disconnected from this sequential, belief-driven process. Most models treat videos as a 'bag of frames', where a subset is uniformly sampled from the video (OpenAI, 2024; Bai et al., 2023; 2025; Cheng et al., 2024; Liu et al., 2023). Lacking an evolving belief about the video's story, uniform sampling is much more likely to sample highly frequent mundane moments over rare surprising (and therefore memorable) events. This can potentially overwhelm Video-LLMs with redundant information, over pivotal moments a human observer would focus on, such as the fall in Figure 1.

To overcome this, some methods select or retrieve frames retroactively for a given textual query (Yu et al., 2025; Wang et al., 2025; 2024; Liang et al., 2024; Tang et al., 2025b). However, in dynamic, open-world settings, we often don't know in advance what questions will be asked. What we need instead is a model that reasons *proactively*, anticipating what is surprising, and paying attention to these shifts, similar to a human observer. In this work, we study two fundamental questions to bridge this gap: (1) How can Video-LLMs proactively track and update their beliefs as new visual evidence presents itself? and (2) Can detecting semantically surprising events proactively and ahead of downstream queries improve video understanding?

---

[1]Code, data and models will be made public at https://github.com/sahithyaravi/SPIKE-RL..

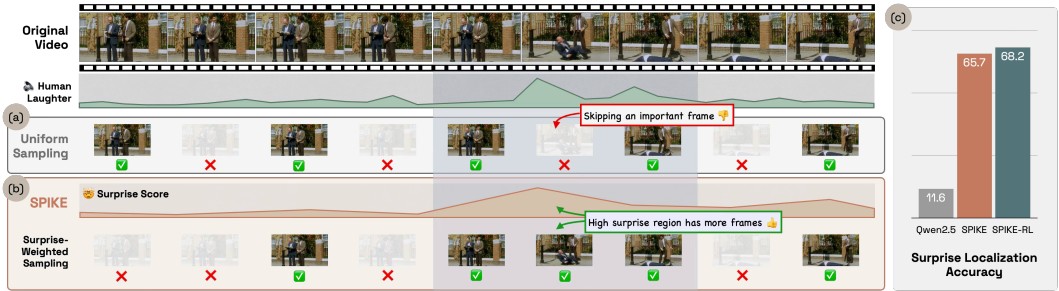

Figure 1: (a) Uniform sampling misses key moments. (b) Our surprise-based sampling focuses on high-surprise regions, strongly aligning with human laughter. (c) Our method achieves significantly better surprise localization than a zero-shot Qwen2.5-VL baseline.

To answer these, we introduce SPIKE, an inference-time framework that represents a model's beliefs as explicit probability distributions over human-interpretable textual hypotheses, and quantifies Bayesian Surprise as the divergence between prior and posterior beliefs (Itti & Baldi, 2005), giving us a surprise score. As shown in Figure 1(b), this surprise score pinpoints moments that contradict the model's prior beliefs. We further improve the surprise scoring by introducing SPIKE-RL, trained using a reinforcement learning objective that teaches the model to prioritize beliefs that lead to more more accurate video captions. SPIKE achieves 65.7% on FunQA (Xie et al., 2025), a surprise localization benchmark, and SPIKE-RL improves on it further, with 68.2%, significantly outperforming the zero-shot performance of Qwen2.5-VL (Figure 1(c)). Our experiments show that SPIKE-RL delivers two complementary benefits: it improves the diversity of generated belief hypotheses, and boosts surprise localization accuracy beyond what the inference-time scorer alone can achieve. Finally, we leverage this signal by replacing the standard uniform frame sampling with surprise-weighted sampling in Qwen2.5-VL and demonstrate that this leads to consistent improvements on five downstream video understanding tasks.

Our approaches allow Video-LLMs to focus on the most salient parts of the video, akin to human notions of surprise. In the future, surprise-aware Video-LLMs can be used to improve the robustness real-time applications such as streaming, surveillance, robotics, and interactive agents that need to adapt to new information on-the-fly.

## 2 BAYESIAN BELIEF TRACKING

### 2.1 SURPRISE SCORING

The architecture of SPIKE is shown in Figure 2. SPIKE quantifies Bayesian surprise by tracking how the model's belief distribution over human-interpretable textual hypotheses shifts when a new frame is observed. Each incoming frame updates this belief distribution, and the magnitude of the change defines the surprise score. SPIKE produces surprise scores for each step, across the complete video. For simplicity, we describe this process using fixed-length videos. However, our method can be adapted to a streaming video setup by applying the same update online.

**Setup.** A video is composed of a sequence of frames $X_{1:T}$, where $T$ is the length of the video. To compute surprise at a timestep $t$, we use three key inputs as shown in Figure 2(b): (i) the *prior window* of $W$ frames immediately preceding the current $t$, $\mathcal{W}_t = X_{t-W:t-1}$, (ii) a *historical summary*, $H_t$, a textual summary of what happened so far in the video, derived from the $C$ frames, $X_{t-C:t-W-1}$, that occurred before $\mathcal{W}_t$,[2] and (iii) the newly observed frame $O_t = X_t$. This setup allows the model to form beliefs based on both long-term context and recent events, and then measure surprise with respect to the new information.[3]

**Hypothesis Generation.** First, at timestep $t$, we generate a set of belief hypotheses, $\mathcal{B}_t = \{b_{t,1}, \ldots, b_{t,N}\}$, where each hypothesis $b$ is a textual description of what might happen next, gen-

---

[2]See Appendix B for further information on how the textual summary is obtained.
[3]See Appendix A.1 for the prompts for the hypothesis generation and scoring.

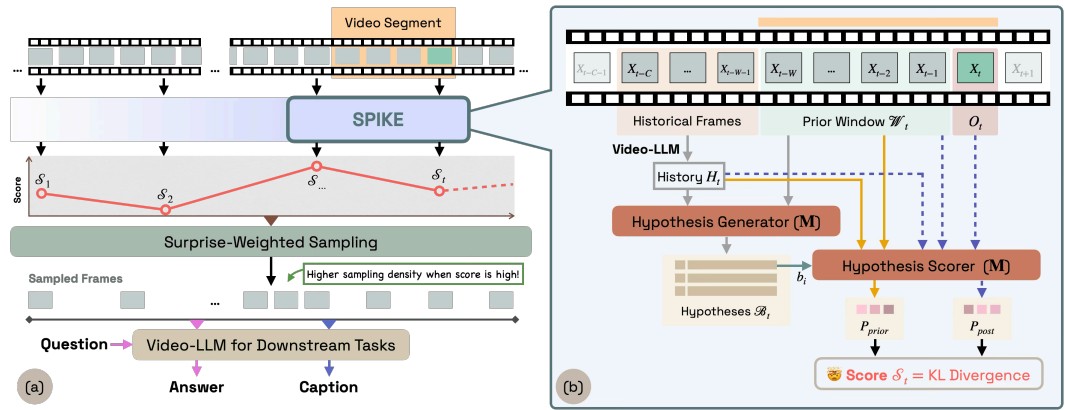

Figure 2: (a) Overall architecture: SPIKE computes surprise scores, which guide weighted frame sampling for downstream tasks. (b) SPIKE : Given history $H_t$, prior window $W$, and observed frame $O_t$, the hypothesis generator produces belief set $B_t$. The hypothesis scorer computes $P_{prior}$ and $P_{post}$, yielding surprise score $S_t$ as KL divergence.

erated by a model $\mathbf{M}$ by conditioning on the historical summary $H_t$ and the prior frame window $\mathcal{W}_t$ (Fig. 2). We use a Video-LLM as our model $\mathbf{M}$ and generate diverse beliefs $\mathcal{B}_t$ using nucleus sampling (Holtzman et al., 2020).

**Bayesian Surprise.** Next, we establish **prior** and **posterior** belief distributions over the generated beliefs $\mathcal{B}_t$. We define a score for each hypothesis $b_{t,i}$ based on its plausibility, which is inversely proportional to its negative log-likelihood (NLL) as computed by the Video-LLM $\mathbf{M}$. This score reflects how well the hypothesis aligns with the given context.

The prior distribution $P_{\text{prior}}$ is calculated based on the historical context ($H_t$) and the recent prior window ($\mathcal{W}_t$), *before* the new frame $O_t$ is observed:

$$P_{\text{prior}}(b_{t,i} \mid H_t, \mathcal{W}_t) = \frac{\exp\left(-\frac{1}{\tau} \cdot \text{NLL}(b_{t,i} \mid H_t, \mathcal{W}_t)\right)}{\sum_{j=1}^{N} \exp\left(-\frac{1}{\tau} \cdot \text{NLL}(b_{t,j} \mid H_t, \mathcal{W}_t)\right)}, \tag{1}$$

where $\text{NLL}(b_i \mid \cdot) = -\log P_{\mathbf{M}}(b_i \mid \cdot)$ is the negative log-likelihood of the hypothesis tokens given the context, and $\tau$ is a temperature parameter. We apply softmax to normalize the scores into a probability distribution.

After observing the new frame $O_t$, we update our beliefs to form the posterior belief distribution, $P_{\text{post}}$, by incorporating this new visual evidence into the model's context:

$$P_{\text{post}}(b_{t,i} \mid H_t, \mathcal{W}_t, O_t) = \frac{\exp\left(-\frac{1}{\tau} \cdot \text{NLL}(b_{t,i} \mid H_t, \mathcal{W}_t, O_t)\right)}{\sum_{j=1}^{N} \exp\left(-\frac{1}{\tau} \cdot \text{NLL}(b_{t,j} \mid H_t, \mathcal{W}_t, O_t)\right)}. \tag{2}$$

Following the Bayesian formalization of surprise by Itti & Baldi (2005), we quantify our surprise score to be the information gain induced by $O_t$, as the Kullback–Leibler (KL) divergence between posterior and prior beliefs over hypotheses:

$$\mathcal{S}_t = D_{\text{KL}}\left(P_{\text{post}}(\cdot \mid H_t, \mathcal{W}_t, O_t) \,\big\|\, P_{\text{prior}}(\cdot \mid H_t, \mathcal{W}_t)\right) \tag{3}$$

$$= \sum_{i=1}^{N} P_{\text{post}}(b_{t,i}) \, \log \frac{P_{\text{post}}(b_{t,i})}{P_{\text{prior}}(b_{t,i})}. \tag{4}$$

Using Equation 3, at each timestep $t$ we compute a scalar surprise score $\mathcal{S}_t$, as well as a belief set at $t$ containing hypotheses and their prior and posterior probabilities, $\mathcal{B}_t = \{(b_{t,i}, P_{\text{prior}}(b_{t,i}), P_{\text{post}}(b_{t,i}))_{i=1}^{N}\}_t$. $\mathcal{B}_t$ is human-readable and interpretable, enabling insight into *why* a video segment is surprising.

## 2.2 SURPRISE-WEIGHTED FRAME SAMPLING

Since it is computationally infeasible and impractical to process all frames of a video, Video-LLMs sample frames – by default, uniformly. Only the selected frames are then processed by the model while the rest are discarded. We define frame budget, $F$, as the maximum number of frames that a Video-LLM uses. Our goal is to effectively select those $F$ frames among the video frames $X_{1:T}$ by recognizing surprising regions of the video, which may be especially important for downstream tasks such as captioning and question answering.

**Computing a Surprise-Guided Probability Distribution.** As shown in Fig 2(a), for a given video $X_{1:T}$, we first uniformly sample timesteps $t_1, \ldots, t_K$, for $K \leq F$. Each timestep represents the end of a video segment, on which we measure surprise; this is akin to a sliding window over the frames of the video. We use SPIKE to compute surprise scores for each segment, and obtain scores $\mathcal{S}_1, \ldots, \mathcal{S}_K \in [0, 1]$ for the corresponding timesteps $t_1, \ldots, t_K$. We can now modify the frame sampling to be proportional to the surprise scores. Specifically, we compute the probability of sampling from a segment ending at $t_i$ as the softmax over scores, $p_i = \text{softmax}\left(\frac{s_i}{\tau_s}\right) = \frac{\exp(s_i/\tau_s)}{\sum_{j=1}^{K} \exp(s_j/\tau_s)}$ $(\tau > 0)$, and use $p_i{=}1/K$ if all $s_i$ are equal. $\tau_s$ is the temperature of this softmax function.

**Sampling.** Given the frame budget $F$ for the Video-LLM, we sample $F$ frames by repeatedly choosing a segment $i$ with probability $p_i$ (with replacement) and drawing a uniform timestamp within that segment; each timestamp is mapped to a frame index via the video frame rate. Choices are independent, so high-surprise segments can contribute multiple frames. We use $\tau_s$ in Eq. 2.2 to control sampling: a small $\tau_s$ concentrates the budget on surprising regions, whereas a larger $\tau_s$ spreads the frame budget more uniformly. We set $\tau_s = 0.7$ for our experiments.

## 2.3 COMPLEXITY ANALYSIS

Let a video contain $T$ frames. We uniformly sample a fixed budget of $F$ frames, so the video is divided into $W = T/F$ segments and one frame is drawn from each segment. For each sampled frame we generate $N$ text hypotheses and compute their prior and posterior likelihoods.

**Time Complexity.** The method requires $F$ hypothesis-generation steps and two batched likelihood evaluations per step. The total cost is therefore $O(F \cdot N)$, which is linear in the chosen frame budget $F$ (and therefore at most linear in $T$ if $F$ grows with $T$). In practice, GPU parallelization allows batching the $N$ hypotheses at each step, amortizing the generation cost and reducing the total complexity from $O(F \cdot N)$ to $O(F)$ when sufficient parallel compute is available.

**Relation to Inference-Time Scaling.** Our overhead is comparable to recent inference-time scaling methods for Video-LLMs: a controllable number of extra forward passes improves where the model allocates its fixed frame budget, without changing its architecture.

**Interpretability.** Because SPIKE represents beliefs as *textual hypotheses*, its Bayesian surprise scores are interpretable: one can inspect the generated hypotheses to understand what the model "expected" versus what the new frames revealed.

## 3 REINFORCEMENT LEARNING FOR BELIEF OPTIMIZATION

**Motivation.** The effectiveness of SPIKE relies on the model's ability to generate belief hypotheses that are accurate, diverse, and representative of the video segment shown. However, since VLMs, are not tailored to perform belief tracking on frame windows, the model has no incentive to refine its intermediate hypotheses. However, training SPIKE with direct supervision on this reasoning process is intractable, as it is impractical to collect ground truth hypotheses across every segment of a video, for a large set of videos. Instead, we leverage GRPO (Shao et al., 2024) to optimize SPIKE using reinforcement learning. SPIKE-RL is based on the insight that a strong final caption – i.e. of what happened in the complete video – is built upon accurate intermediate belief hypotheses – i.e. about what is likely to happen after having watched a portion of the video.

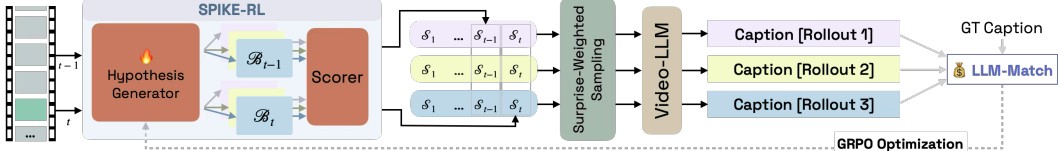

Figure 3: SPIKE-RL explores multiple hypothesis trajectories, whose surprise scores guide frame sampling. Captions from these rollouts are scored with LLM-Match, and GRPO propagates the reward to improve hypothesis generation.

Figure 3 demonstrates our approach. To train the hypothesis generator, our policy model, we compute a reward signal based on the quality of the final caption. This reward signal is then propagated backward, assigning credit to the sequence of beliefs that led to the successful outcome. In this way, supervision on the final result is implicitly transformed into training feedback for the model's internal reasoning process. Our rewards are derived from an LLM-based metric that computes the similarity between the generated caption and the ground truth caption.

**Rollout.** We design the GRPO-based training procedure by generating a group of captions, based on different *trajectories* of beliefs and frame allocations. For each video, we draw $M$ *trajectories* $\{\tau^{(r)}\}_{r=1}^M$. Each trajectory $\tau^{(r)}$ runs SPIKE over segments of the video. At every timestep $t$, it samples $N$ textual beliefs $\mathcal{B}_t^{(r)} = \{b_{t,1}^{(r)}, \ldots, b_{t,N}^{(r)}\}$ and scores prior and posterior beliefs to obtain $(P_{\text{prior},t}^{(r)}, P_{\text{post},t}^{(r)})$ and the surprise scores $\mathcal{S}_t^{(r)}$. We then use the surprise scores to inform the sampling of frames that are inputted into a Video-LLM to generate a single final video caption, $c^{(r)}$ using our surprise-based frame allocation (§2.2). Thus each input induces a GRPO *group*:
$\mathcal{G} = \left\{ \left( \{\mathcal{B}_t^{(r)}, \mathcal{S}_t^{(r)}\}_t^T, c^{(r)} \right) \right\}_{r=1}^M$

**Reward.** At the end of a rollout, the caption $c^{(r)}$ is scored using LLM-Match, where an LLM judge measures how similar it is to the ground truth caption, to obtain a scalar reward $R^{(r)}$. The prompt for the LLM judge is in Appendix A.2. We Z-score the LLM rewards within the group, and use the normalized scores as advantages in the policy objective, $A^{(r)} = \frac{R^{(r)} - \mu_R}{\sigma_R}$.

**Loss.** We treat the full set of hypotheses in a trajectory as a sequence-level action. Let $p_\theta(b_{t,k} \mid H_t, \mathcal{W}_t)$ denote the policy for generating a hypothesis given the video context. We define our **belief-optimization** objective as,

$$\mathcal{L}_{\text{belief-optimization}}(\theta) = -\frac{1}{M} \sum_{r=1}^M A^{(r)} \left( \sum_t \sum_{k=1}^K \log p_\theta(b_{t,k}^{(r)} \mid H_t^{(r)}, \mathcal{W}_t^{(r)}) \right), \tag{5}$$

which increases the likelihood of hypotheses along high-advantage trajectories and suppresses those along low-advantage ones. Optimizing Equation 5 trains the model to generate hypotheses that reliably support strong captions, improving both the intermediate belief trajectory and the final output.

**Training.** For training SPIKE-RL, we curated a video captioning dataset of 2,000 videos with 30% *surprising* and 70% *unsurprising* videos. The goal is to expose the policy both to routine events where beliefs remain stable and to inflection points that induce belief shifts. For the unsurprising portion, we used ActivityNet Captions (Caba Heilbron et al., 2015), which predominantly includes videos depicting everyday activities. For the surprising videos, we sample from from the training set of Oops! (Epstein et al., 2020), a collection of short clips centered on unintentional human failures. We use `Qwen2.5-VL-7B-Instruct` as the Video-LLM model (**M**) and `Olmo-7B-hf` as the LLM-Match reward model. See App. C for the training hyperparameters.

## 4 SURPRISE LOCALIZATION

We first evaluate how well SPIKE and SPIKE-RL can identify surprising segments of a video. Hyperparameters for surprise scoring are described in App. C.

## 4.1 EXPERIMENTAL SETUP

**Benchmarks.** We evaluate surprise localization on three benchmarks: Oops! (Epstein et al., 2020), FunQA (Xie et al., 2025) and Mr. Bean (App. E). Oops! is a surprise detection task, whose test set contains 4,791 videos with precise timestamps marking the exact transition point to surprise. FunQA has 424 videos with annotations for the most surprising segment in each video, given by a start and end time. While these are established benchmarks, they only annotate a single surprising event per video. Since our method is capable of detecting multiple surprising segments in the video, we curate our own benchmark, Mr. Bean, using 48 clips from the live-action TV show. Mr. Bean's audio laughter track serves as silver-standard surprise annotations – segments of the video with laughter are considered surprising.

**Metrics.** Following the protocols of Oops! and FunQA, we report Acc@0.25s and Acc@1.0s for Oops!, and IoU for FunQA. The accuracy metrics (Acc) measure whether the predicted surprise peak falls within 0.25 or 1.0 seconds of the ground truth peak surprise, while IoU measures the overlap between the predicted surprising windows and the ground-truth surprising windows. For details on the implementation of the metrics, see App. D.

**Baselines.** We establish a lower bound with a Random baseline that selects surprising frames at random. We also report the zero-shot performance of our base `Qwen2.5-VL-7B-Instruct` model, which directly scores each uniformly sampled frame on whether it is surprising or not, without our proposed belief tracking mechanism (See Appendix A.3 for the prompt and setup). On Oops!, we compare against: (i) VideoSpeed (Epstein et al., 2020), the strongest reported baseline for this dataset; (ii) Motion Magnitude (Epstein et al., 2020), an optical-flow-based approach; and (iii) F2C2V (Duka et al., 2022), a self-supervised method. As an upper-bound reference, we also report the human consistency or agreement from the original dataset. On FunQA, we compare against TimeChat (Ren et al., 2023), UniVTG (Lin et al., 2023), a specialized video temporal grounding framework, and LLaVA-Next-CR, a baseline provided by the FunQA benchmark that applies the clipping and rating (CR) technique from UniVTG to LLaVA-NeXT (Liu et al., 2024).

## 4.2 RESULTS

Table 1 shows the performance of SPIKE and SPIKE-RL on the surprise localization task. On the Oops! benchmark, our SPIKE-RL model achieves an score of 62.9% on Acc@0.25s, remarkably close to the human performance (62.1%). Notably, both SPIKE and SPIKE-RL show about a tenfold improvement over the performance of the zero-shot version of the same model (`Qwen2.5-VL-7B`). Compared to VideoSpeed, F2C2V, we observe that SPIKE and SPIKE-RL are better at accurate localization, with a 23.4% higher Acc@0.25s, and achieve similar Acc@1s scores. On the FunQA benchmark, SPIKE-RL once again demonstrates superior performance with an IoU of 68.2, surpassing both prior approaches and the zero-shot model by a substantial margin. It is worth noting that this significant boost is despite the fact that FunQA – which is composed of positive surprises related to humor and creativity – is out-of-distribution for SPIKE-RL.

Mr. Bean shows a similar trend to the other benchmarks, but the absolute Acc@0.25s is lower. This dataset is particularly challenging. In contrast to the other benchmarks, some of the surprising moments in Mr. Bean arise from subtle, fine-grained nuances in his facial expressions rather than easily noticeable unexpected events. Finally, we observe a significant 6.3% gain in IoU score with SPIKE-RL over SPIKE. Since IoU on Mr. Bean evaluates detection across multiple surprising segments, this gain highlights the ability of our scorer to capture nuanced surprises within a video.

Overall, the inference-time method, SPIKE, achieves superior performance across all benchmarks and generalizes to different types of surprises, while SPIKE-RL further boosts performance through reinforcement-guided refinement.

## 4.3 BELIEF SET EVALUATION

We evaluate the hypotheses generated by SPIKE and SPIKE-RL using a combination of automatic metrics and human evaluation.

**Diversity.** We are interested in whether models generate multiple conceptually-diverse hypotheses or different lexical variations of the same hypothesis. For a given video, we measure the diversity

Table 1: Performance of SPIKE and SPIKE-RL on surprise localization.

| Method | Oops! | | FunQA | Mr. Bean | | |
| | Acc@0.25s | Acc@1s | IoU | Acc@0.25s | Acc@1s | IoU |
|---|---|---|---|---|---|---|
| *Baselines* | | | | | | |
| Random | 6.8 | 2.6 | 7.5 | 0.6 | 3.5 | 0.9 |
| Motion | 23.1 | 50.7 | – | – | – | – |
| Video Speed | 36.6 | 65.3 | – | – | – | – |
| F2C2V | 39.5 | **69.5** | – | – | – | – |
| TimeChat | – | – | 9.6 | – | – | – |
| UniVTG | – | – | 45.3 | – | – | – |
| LLaVA-NeXT-CR | – | – | 62.3 | – | – | – |
| Qwen2.5-VL | 6.6 | 9.6 | 11.6 | 11.2 | 23.2 | 13.8 |
| SPIKE | 60.0 | 67.3 | 65.7 | 53.2 | 70.2 | 54.8 |
| SPIKE-RL | **62.9** | 69.1 | **68.2** | **57.4** | **78.7** | **61.1** |
| Human | 62.1 | 88.0 | – | – | – | – |

Table 2: Performance of Qwen2.5-VL with uniform vs. surprise-weighted and other query-free frame sampling methods. MCQ tasks are evaluated with accuracy; generative tasks with LLM-Match. Comparable open-source Video-LLMs are shown for context.

| Model | Size | Sampling | BlackSwan | FunQA | ExFunTube | VideoMME-S | NextQA |
|---|---|---|---|---|---|---|---|
| VideoChat2 | 7B | Uniform | 49.7 | 17.9 | – | 45.6 | – |
| VideoLlama2 | 7B | Uniform | 52.9 | 7.7 | – | 56.0 | – |
| FunMentor | 7B | Uniform | – | 33.2 | – | – | – |
| LLaVA-Video | 7B | Uniform | 70.4 | – | – | 46.6 | 62.7 |
| Qwen2.5-VL | 7B | Uniform | 67.2 | 66.8 | 68.7 | 59.8 | 68.6 |
| Qwen2.5-VL | 7B | RGB Histogram | 49.6 | – | – | 55.4 | – |
| Qwen2.5-VL | 7B | ECR | 49.7 | – | – | 58.2 | – |
| Qwen2.5-VL | 7B | Katna | 54.6 | – | – | 57.4 | – |
| Qwen2.5-VL | 7B | Optical Flow | 58.6 | – | – | 58.1 | – |
| Qwen2.5-VL | 7B | SPIKE | 68.8 | 70.3 | 73.2 | 60.8 | 69.8 |
| Qwen2.5-VL | 7B | SPIKE-RL | 69.5 | 71.4 | 75.7 | 62.5 | 70.3 |
| Qwen2.5-VL | 32B | Uniform | 69.4 | 72.7 | 71.9 | 69.9 | 72.3 |
| Qwen2.5-VL | 32B | SPIKE-RL | 71.7 | 75.8 | 75.8 | 73.5 | 74.1 |

of a hypothesis set with the average inverse cosine similarity $(1 - cos(b_i, b_j))$ across all hypothesis pairs. SPIKE-RL achieves 40.3%, higher than SPIKE's 33.5%, showing that the RL training improves diversity.

**Correlation with human judgments.** We measure how well our surprise score aligns with human judgments by showing human annotators a random sample of 100 videos from Oops! along with the generated hypotheses and asking them to rank the hypotheses by dragging them onto a 0–100 scale. Each video segment is evaluated twice: first using only the prior frames ($O_{<t}$), and then again after revealing the observed frame ($O_t$). This setup emulates the prior and posterior probabilities in Eq. 3, from which we compute human-derived surprise scores. Comparing these to SPIKE and SPIKE-RL's surprise scores yields a Spearman correlation of 0.84 and 0.87 respectively, indicating **very strong correlation** and confirming that our method effectively captures belief shifts. The template for human evaluation is provided in App. H.

## 5 DOWNSTREAM TASKS

Having shown that SPIKE and SPIKE-RL can perform surprise localization, we now explore how identifying surprising segments of the video and allocating more frames to such regions can improve a Video-LLM's performance on downstream tasks as described in §2.2.

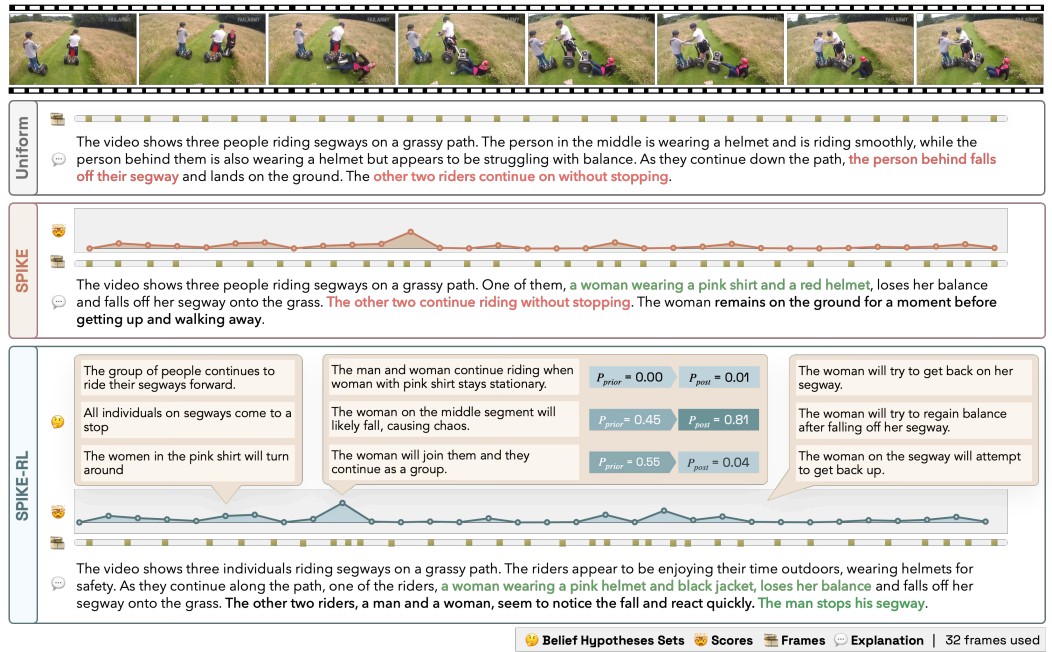

Figure 4: **Qualitative Results.** We compare Uniform, SPIKE and SPIKE-RL sampling methods. Errors in the explanation generated using uniform sampling reduce with SPIKE and are resolved with SPIKE-RL. We show belief hypotheses sets ($\mathcal{B}_t$) at various timesteps, and observe how the divergence of $P_{prior}$ and $P_{post}$ accurately captures the surprising moment in the video.

## 5.1 EXPERIMENTAL SETUP

**Benchmarks.** We evaluate our sampling method on a diverse selection of tasks, spanning surprise explanations, question answering, and temporal reasoning. The Reporter-MCQ portion of Black-SwanSuite (Chinchure et al., 2025) tests models' ability to describe an unexpected event in a MCQ setup. FunQA's Task 2 (Xie et al., 2025) and ExFunTube (Dayoon Ko, 2023) ask models to generate an explanation of why events are surprising. Moving beyond surprising videos, we test our models on two MCQ tasks – VideoMME (Fu et al., 2024), which probes general multimodal reasoning (we focus on short videos without subtitles), and NextQA (Xiao et al., 2021), which targets temporal, commonsense, and causal reasoning.

**Metrics.** Following prior work (Majumdar et al., 2024; Xie et al., 2025), we evaluate the generative tasks using LLM-Match, prompting GPT-4o to rate the similarity between model-generated and ground-truth answers. Multiple-choice tasks are evaluated using accuracy.

**Video-LLM Baselines**. We consider widely adopted open-source Video-LLMs capable of video explanation and QA, including VideoChat2 (Li et al., 2024), VideoLlama2 (Cheng et al., 2024), and LLaVA-Video (Liu et al., 2023). We also include FunMentor (Xie et al., 2025), a model specifically designed for humor understanding. Our base model is Qwen2.5-VL (7B), which we use to evaluate alternative sampling strategies under a fixed frame budget on BlackSwan and VideoMME-S. Finally, we test whether SPIKE-RL improves performance on a larger model, Qwen2.5-VL (32B).

**Query-free Frame Sampling Baselines**. To assess the effectiveness of our sampling, we benchmark against shot boundary detection methods on BlackSwan and Video-MME-S. Specifically, we tested RGB Histogram differences (V & Narayanan, 2015), Edge Change Ratio (ECR; Mann & Kaur, 2015), and motion-based detection (Wolf, 1996), which capture changes in texture, structure, motion, and similarity. In all of these approaches, salient peaks are detected via smoothed scores and frames are distributed proportionally to peak strength, ensuring that the frame budget $F$ is met. We also include Katna,[4] a clustering-based method which applies K-means to frame histograms and

---

[4] https://github.com/keplerlab/katna

selects the frame closest to each centroid. We use a maximum frame budget $F$ of 64 frames for all our baselines, regardless of the sampling approach.

## 5.2 RESULTS

Table 2 shows the performance of SPIKE and SPIKE-RL on downstream benchmarks. On tasks with surprising videos (BlackSwan, FunQA, ExFunTube), surprise-aware sampling provides substantial gains over uniform selection. Relative to uniform sampling, SPIKE improves accuracy by +1.6% on BlackSwan, +3.5% on FunQA, and +4.5% on ExFunTube. We observe that SPIKE-RL further extends performance on these tasks, with gains of +2.3% and +4.6% on BlackSwan and FunQA, and +7.0% on ExFunTube, marking our largest gains over uniform sampling. These results not only show the effectiveness of SPIKE in prioritizing surprising frames, but also credit the improved hypothesis quality in SPIKE-RL. On Qwen2.5-VL 32B, we see 2.3%, 3.1% and 3.9% gains respectively with SPIKE-RL, showing that our methods benefit larger models as well, extending their video understanding capability.

In general QA tasks (VideoMME-S, NextQA), we see moderate but consistent improvements over uniform sampling. SPIKE boosts scores by +1.0% on VideoMME-S and +1.2% on NextQA, while SPIKE-RL achieves +2.7% and +1.7% respectively on the 7B variant. The 32B variant with SPIKE-RL shows larger improvements of 3.6% and 1.8% on these tasks. These results show that surprise-aware sampling is broadly beneficial.

SBD strategies such as RGB Histogram, ECR, Katna, and Optical Flow consistently underperform uniform sampling. Their reliance on raw visual change makes them sensitive to camera motion and scene cuts, which rarely align with semantically important events. In contrast, our method offers principled guidance for identifying critical moments. Overall, we demonstrate that Bayesian Surprise provides a powerful inductive signal for adaptive frame selection: SPIKE delivers immediate gains by reallocating a fixed frame budget toward more informative segments, while SPIKE-RL further improves robustness through reinforcement-guided belief optimization.

## 5.3 QUALITATIVE EXAMPLE

Figure 4 illustrates the differences between uniform sampling, SPIKE, and SPIKE-RL. Under uniform sampling, the Video-LLM generates a caption that notes someone falling off a segway but misidentifies the person and the actions of the other riders (error highlighted in red). With the same frame budget, SPIKE and SPIKE-RL reallocate samples toward segments with high surprise scores, guided by observed belief shifts as demonsrated by the hypotheses. SPIKE correctly captures that the woman in the pink shirt and helmet loses balance and falls, though it still makes an error by stating that the other riders continue without stopping. SPIKE-RL improves on this. By more accurately localizing surprising segments – with one peak at the main fall and another smaller peak later – SPIKE-RL increases sampling density around both critical events. This leads to a more precise description of both the fall and the subsequent reactions of the other riders.

## 6 RELATED WORK

**Belief Tracking and Updating.** Recent research in NLP has explored the idea of maintaining and updating beliefs, often with Bayesian inspired methods. Studies show that, with sufficient evidence, LLMs can approximate Bayesian updates rather than simply relying on pattern matching (Gupta et al., 2025). Closest to our work, Kim et al. (2025) explicitly maintain and re-weight hypotheses about agents' mental states as new information becomes available, mirroring Bayesian Theory of Mind. This principle of explicit tracking also improves model robustness in complex scenarios involving multiple characters and higher-order Theory of Mind (Sclar et al., 2023). This process is closely related to the concept of defeasible reasoning, where conclusions are revised by new evidence (Rudinger et al., 2020). More broadly, the principle of Bayesian Surprise has been used as a powerful driver for exploration in other domains, such as guiding open-ended scientific discovery (Agarwal et al., 2025). We extend this idea of discovery to the domain of video understanding.

**Adaptive Frame Sampling Strategies for Video-LLMs.** Prior work on frame selection for Video-LLMs is primarily based on the relevance to the question. Query-conditioned methods rank frames

with respect to a textual prompt to minimize redundancy while preserving task-relevant evidence. Frame-Voyager(Yu et al., 2025), Flexible Frame Selection (FFS; Buch et al., 2025) and Hu et al. (2025) learn to select informative frame sets conditioned on the query using lightweight training strategies. Adaptive Keyframe Sampling (AKS; Tang et al., 2025b) formulates selection as a plug-and-play module optimizing relevance to the prompt and Guo et al. (2025) propose dynamic keyframe search driven by visual chain-of-thought. VideoTree (Wang et al., 2025) organize a video into a hierarchical tree and traverse it in a question adaptive manner. In contrast, our method drops in as a replacement for the Video-LLM's uniform sampling layer, reallocating the frame budget towards surprising moments while remaining query-agnostic.

**Video Saliency and Attention.** Several works have explored video saliency and attention mechanisms to focus on important frames. Hu et al. (2025) uses differential keyframe selection to choose salient frames and differential feature merging to compress non-keyframes, focusing on query-relevant information while reducing redundancy, thus improving long-form QA. Ma et al. (2025) introduces Video Token Sparsification (VTS) is a CNN-based approach to reduce visual tokens for efficient MLLM use in autonomous driving. Lee et al. (2025) introduces LLMVS, a video summarization framework using LLMs to evaluate frame importance based on captions and refines it based on global attention mechanism. Tang et al. (2025a) introduces Adaptive Keyframe Sampling (AKS), that performs keyframe selection to maximize useful information within token limits, optimizing for relevance to the prompt and coverage of the video. Azad et al. (2025) introduces HierarQ, a hierarchical Q-Former framework that processes video frames sequentially using short and long-term memory banks for enhanced temporal modeling and task-aware video comprehension. Koala Tan et al. (2024) is a key frame-conditioned long Video-LLM, which uses learnable spatiotemporal queries to adapt pretrained VLMs for longer videos. The key distinction in our work is that we explictly peform belief tracking, and show that our surprise is correalted with humans. Frame selection is our downstream application, but the belief tracking process could be useful for explainable reasoning, video streaming, and training models with grounded belief trajectories in the future.

**Bayesian Theory of Mind and Prediction Error.** Bayesian models of social cognition frame human reasoning about others through Bayesian Theory of Mind (BToM), where observers infer latent beliefs and goals by inverting a generative model of action Baker et al. (2009; 2017). Complementing these Bayesian approaches, predictive coding theory proposes that the brain continuously generates predictions about sensory input and uses prediction errors to update internal models Rao & Ballard (1999); Millidge et al. (2022). Empirically Koster-Hale & Saxe (2013), extended predictive coding to theory of mind, with brain activity showing reduced responses for predictable versus unpredictable beliefs. In this integrated view Thornton et al. (2018), BToM provides the content of the generative model (e.g., "She is thirsty"), while predictive coding describes the process of maintaining and updating this model. SPIKE follows a similar approach of maintaining futures and updating its belief scores.

## 7 CONCLUSION

We introduced SPIKE, a framework that lets Video-LLMs proactively register surprise. We further showed that SPIKE-RL can refine intermediate belief generation, improving both belief diversity and surprise localization. This enables surprise-driven frame sampling, yielding consistent gains across downstream tasks, especially when critical information is sparse. Modeling surprise offers a path toward understanding video narratives, adapting to violated expectations, and anticipating events. Future work could investigate extending SPIKE to real-time streams and combining with task-specific relevance signals.

## 8 REPRODUCIBILITY STATEMENT

We intend to make all our data, code and models open-source. SPIKE is based on an open source Video-LLM, Qwen2.5-VL, and our training code for SPIKE-RL will be made available on GitHub. We note that all our prompts are included in Appendix A and hyperparameters in Appendix C. For the Mr. Bean evaluation set we create, we plan to share the video clips, along with annotations containing their original source. We also share the instructions and template used in our human evaluation in Appendix H.

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

# A   PROMPTS

## A.1   HYPOTHESIS PROMPTS

**Generation.** We prompt the model with a memory of prior events and recent frames, asking for a concise next–frame prediction (8–10 words):

> Given a textual summary of the video so far and the most recent *prior window of frames*, predict what will most likely happen in the next frame.
> **Context so far:** memory_text
> **Prior window (video inputs):** *A sequence of images corresponding to the last W frames.*
> **Output format:** Hypothesis: 8–10 words

**Prior.** We use the following prompt to score each hypothesis.

> **Context so far:** memory_text
> **Prior window (video inputs):** A sequence of images corresponding to the last $W$ frames.
> **Current frame:** The observed frame immediately following the prior window.
> Here is what will happen next: [hypothesis statement]

**Posterior.** We use the following prompt to score each hypothesis and compute the probability of yes as the posterior likelihood of that hypothesis.

> You are given a textual summary of the video so far, a *prior window* of frames, and the *current frame* that follows. Your task is to evaluate whether each hypothesis generated from the prior context still holds in the current frame.
> **Context so far:** memory_text
> **Prior window (video inputs):** A sequence of images corresponding to the last $W$ frames.
> **Current frame:** The observed frame immediately following the prior window.
> **Hypothesis:** [hypothesis statement]
> Question: Is this hypothesis true in the *current frame*? Answer with a single word: yes or no.

## A.2   LLM REWARD PROMPT

> Rate how closely the content of the prediction matches the content of the reference description in terms of meaning and how well it captures important details regarding events in the video. Ignore the difference in length. Score 0.0-1.0 where:
> 0.0-0.3: Poor match (key details in the reference are missing in the prediction) 0.4-0.6: Moderate match (a few key details in the reference are captured in the prediction) 0.7-0.9: Good match (most key details are present in the prediction) 1.0: Perfect match (all key details in the reference are accurately captured in the prediction) Output only the numerical score (e.g., 0.75).
> **Reference**: gt
> **Response**: response
> Score:

## A.3   ZERO-SHOT SCORER PROMPT

> You are analyzing video frames for surprisingness. For each frame, assign a label of 1 if it is surprising and 0 if it is not.
> 1: surprising content
> 0: expected content
>
> **Video frames**: *Original Video Frames*

## B    HISTORICAL SUMMMARY

In our implementation, the memory of what happened since the beginning of the video i.e the Historical summary, is maintained as a rolling textual summary that updates with each newly observed frame. Before use, the memory is compressed using the BART-Large-CNN summarization model whenever it exceeds approximately 200 word. For each step, the model receives the condensed memory, a short window of prior frames, and the most recent observed frame, and generates a caption describing the new event. This caption is appended to the memory, yielding a continuously updated narrative of "what has happened so far", which is then used for hypothesis generation and surprise computation.

## C    HYPERPARAMETERS

### C.1    TRAINING

We train using 4 H100s on a single node with DeepSpeed ZeRO-3 offload. All runs use Qwen2.5-VL-7B-Instruct as the backbone, with FlashAttention-2, bfloat16 precision, and PEFT enabled.

Table 3: Key hyperparameters for GRPO training.

| Hyperparameter | Value |
|---|---|
| Learning rate | $1 \times 10^{-6}$ |
| GRPO $\beta$ | 0.1 |
| Number of GRPO Rollouts | 3 |
| Number of Hypotheses per window | 3 |
| Max prompt length | 8192 tokens |
| Training samples | 2000 |
| Epochs | 1 |
| Per-device batch size | 1 |
| Effective global batch size | 4 |
| Random seed | 42 |

### C.2    INFERENCE

For both SPIKE and SPIKE-RL, we maintain a hypothesis set $N = 3$ per time step. We use a prior window of $W = 4$ frames, and the frames for surprise scoring are allocated in proportion to the video duration, $F = f(\text{duration})$. Videos up to a minute are assigned a base budget of 8 frames. For longer videos, the budget continues to double with each additional minute.

## D    SURPRISE LOCALIZATION METRICS

**Accuracy@$\delta$.** Let $\hat{t}$ be the predicted time (in seconds) obtained by converting the frame with the highest surprise score to time, and let $t^\star$ be the ground-truth transition time. We use the transition time provided in Oops! directly. For FunQA and Mr.Bean, center of the most surprising window is used as transition time. The instance-level score is

$$\text{Accuracy@}\delta = \mathbb{1}\left[ |\hat{t} - t^\star| \leq \delta \right],$$

and the reported metric is the mean of this indicator over the evaluation videos. Typical choices include $\delta \in \{0.25, 1.0\}$ seconds.

**IoU.** Let $\mathcal{W}_{\text{pred}} = \{[a, b] : s(t) > \tau \text{ for } t \in [a, b]\}$ be the predicted surprising windows and $\mathcal{W}_{\text{gt}}$ be the given set of ground truth surprising windows. The Temporal IoU is:

$$\text{Temporal IoU} = \frac{\text{intersection coverage}}{\text{union coverage}} = \frac{|\bigcup \mathcal{W}_{\text{pred}} \cap \bigcup \mathcal{W}_{\text{gt}}|}{|\bigcup \mathcal{W}_{\text{pred}} \cup \bigcup \mathcal{W}_{\text{gt}}|}$$

where $|\cdot|$ denotes temporal coverage (total duration). We define predicted surprising windows as a set of maximal contiguous intervals where the surprise score exceeds a threshold $\tau = 0.8 \times \max_t s(t)$ for that video.

# E  MR. BEAN

We collect 48 videos from Mr. Bean compilation videos on YouTube. Specifically, we follow this process:

1. Each clip is divided into its scenes using a scene detector model, PySceneDetect, using its ContentDetector[5], with a threshold of 30.

2. Scenes shorter than 12 seconds and longer than 60 seconds are filtered out, to reduce incorrect scene cuts or have videos that are too short for our analysis.

3. We extract the audio from these scenes, and use a laughter segmentation model from Omine et al. (2024) to identify where laughter is present. We filter scenes to obtain only those that have 1 to 3 laughter segments.

4. Because we rely on laughter tracks as our silver-standard surprise annotation, we transcribe the audio in these clips. We use OpenAI's Whisper (Radford et al., 2023), with the *turbo* model. If a clip has too many words in its transcription ($> 8$), it is discarded. Through empirical observation, we found that laughter occurs in small peaks. We ensure that at least one such loud peak ($> -28dB$) of at least 1 second occurs.

5. As a final step, we manually filter through the video set to discard scenes which contain additional noises (e.g. bells) or scenes that are not semantically meaningful (e.g. the opening credits) that may have passed the other filters. This leaves us with 48 video clips.

The full list of clips, a link to their original source, along with video scenes which we use, will be provided with the code and data release.

# F  JSD

For bounded and symmetric reporting, we convert KL to the Jensen–Shannon divergence (JSD), where,

$$\mathcal{S}_t = \mathrm{JSD}(P_{\mathrm{post}}, P_{\mathrm{prior}}) = \tfrac{1}{2}D_{\mathrm{KL}}(P_{\mathrm{post}}\|M) + \tfrac{1}{2}D_{\mathrm{KL}}(P_{\mathrm{prior}}\|M), \tag{6}$$

where $M = \frac{1}{2}(P_{\mathrm{post}} + P_{\mathrm{prior}})$, which maps naturally to $[0, 1]$ after $\log_2$ normalization.

# G  HYPERPARAMETER SENSITIVITY ANALYSIS

## HYPOTHESIS SET SIZE

We fixed our hypothesis set size to $N = 3$, as it worked well for surprise detection across our benchmarks, and the beliefs are restricted to the most likely beliefs given the context of the video. To understand the sensitivity to hypothesis set size, we conducted experiments with varying values of $N$ on the Oops! and Mr. Bean datasets, reported in Tables 4 and 5, respectively.

Table 4: Effect of hypothesis set size $N$ on the Oops! dataset.

| Approach | # Hypotheses ($N$) | Acc@0.25s | Acc@1s |
|---|---|---|---|
| SPIKE | 3 | 60.00 | 67.30 |
| SPIKE | 5 | 60.36 | 67.73 |
| SPIKE | 10 | 60.34 | 67.69 |

---

[5]https://www.scenedetect.com/docs/0.6.1/api/detectors.html

Table 5: Effect of hypothesis set size $N$ on the Mr. Bean dataset.

| Approach | # Hypotheses ($N$) | Acc@0.25s | Acc@1s |
|----------|--------------------|-----------|--------|
| SPIKE | 3 | 53.20 | 70.20 |
| SPIKE | 5 | 53.67 | 70.82 |
| SPIKE | 10 | 53.70 | 70.86 |

As shown in Tables 4 and 5, increasing from $N = 3$ to $N = 5$ hypotheses yields marginal improvements (0.5–0.6% on Acc@1s), while further increasing to $N = 10$ shows diminishing returns. This occurs because the probability mass is already well captured by the top 5 hypotheses, and adding more hypotheses does not meaningfully change the KL divergence. We therefore adopt $N = 3$ as the default for an optimal balance between accuracy and computational efficiency, though $N = 5$ can be used for additional gains in performance and depending on the length and complexity of the video.

SENSITIVITY TO WINDOW SIZE $W$ AND HISTORICAL SUMMARY

We conducted an ablation study to assess the contribution of the historical summary component. Table 6 reports results on the Oops! dataset with and without the historical summary.

Table 6: Ablation of the historical summary component on the Oops! dataset.

| Approach | Acc@0.25s | Acc@1s |
|----------|-----------|--------|
| SPIKE | 60.0 | 67.37 |
| SPIKE w/o Historical Summary | 53.7 | 61.29 |

The results in Table 6 demonstrate that the historical summary is critical for semantic understanding of surprise. To recognize a surprising event such as a fall, the model must first establish the narrative context (e.g., *a person riding a bicycle smoothly*). Without the textual summary, the model relies solely on the visual buffer $W$, which captures motion but lacks the narrative intent that makes KL divergences semantically meaningful. The historical summary enables the model to detect violations of established narrative expectations, contributing substantially to performance.

We further ablate the window size $W$, with results reported in Table 7.

Table 7: Effect of window size $W$ on the Oops! dataset.

| Approach | $W$ | Acc@0.25s | Acc@1s |
|----------|-----|-----------|--------|
| SPIKE | 2 | 59.55 | 67.04 |
| SPIKE | 4 | 60.00 | 67.37 |
| SPIKE | 6 | 59.97 | 67.33 |
| SPIKE | 8 | 59.33 | 67.32 |

As shown in Table 7, performance peaks at $W = 4$ for the Oops! dataset.

FRAME BUDGET AND LATENCY/THROUGHPUT

Our design ensures that SPIKE is practical for real-world applications. All hypothesis generation and scoring operations run in parallel. Compute time scales linearly with video length (number of timesteps), making performance predictable.

To understand the trade-off between frame budget $B$ and surprise localization accuracy, we vary $B$ on the Oops! dataset, with results reported in Table 8.

As shown in Table 8, at $B = 32$ frames (half our default budget), performance drops by approximately 1.1% in Acc@0.25s compared to $B = 64$. At the extremely low budget of $B = 8$, SPIKE

Table 8: Effect of frame budget $B$ on surprise localization accuracy on the Oops! dataset.

| Approach | $B$ | Acc@0.25s | Acc@1s |
|---|---|---|---|
| SPIKE | 8 | 55.60 | 63.76 |
| SPIKE | 16 | 58.44 | 66.41 |
| SPIKE | 32 | 58.91 | 66.83 |
| SPIKE | 64 | 60.00 | 67.37 |
| SPIKE | 128 | 60.25 | 67.65 |

achieves 55.6% Acc@0.25s, corresponding to a ∼4% drop relative to the default. Performance saturates beyond $B = 64$, with $B = 128$ frames yielding only marginal gains (+0.25% Acc@0.25s), suggesting that the model is approaching the performance ceiling on this benchmark.

## H  HUMAN EVALUATION TEMPLATE

Fig  A1 and Fig A2 show the template and instructions used for human evaluation.

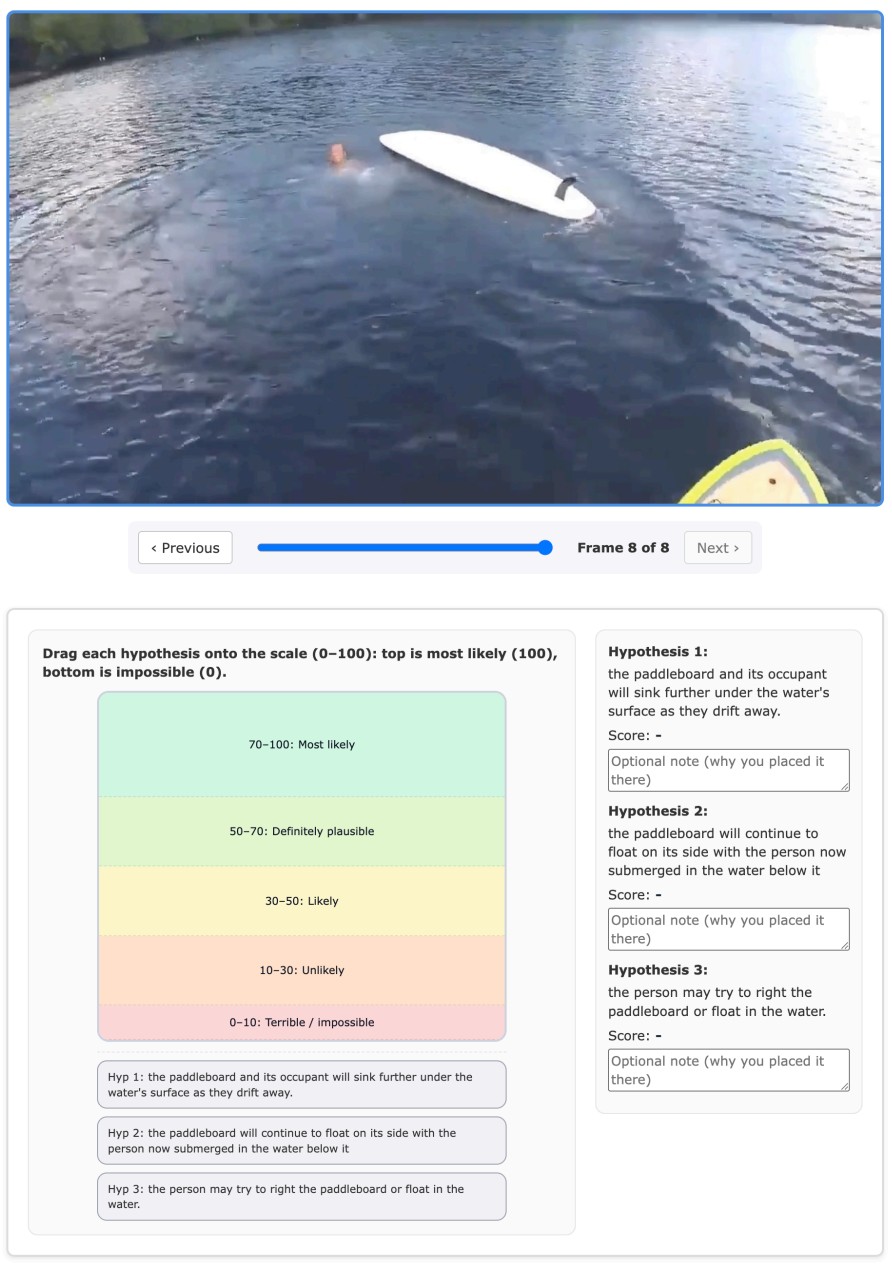

Figure A1: We ask human evaluators to score the hypotheses by dragging and dropping them into likelihood bands between 0 – 100. This is repeated twice – by scoring the hypothesis with and without the observed new frame.

## Task Instructions

**Goal:** You will be shown the frames of a video one after another. For each frame, score each hypothesis based only on what frames have been seen so far.

- Use the slider or Next/Previous to move between frames.

- Only use visual evidence from frames you have seen so far (from the start up to the current frame).

- For each hypothesis shown at the current frame, assign a **likelihood score (0–100)** by dragging Hyp1, Hyp2 and Hyp3 boxes into the colored band shown on the left:
    - **0–10 – Terrible/Impossible**: Contradicted by what you have seen so far.

    - **10–30 – Unlikely**: Little support; seems implausible given the evidence so far.

    - **30–50 – Likely**: Supported by several cues; plausible given the evidence so far.

    - **50–70 – Definitely plausible** to **70–100 – Most likely**: Strongly supported.

- Optional: add a short note explaining why you chose the score.

Figure A2: Instructions shown to human evaluators.

