# OpenReview forum: "SPIKE-RL: Video-LLMs meet Bayesian Surprise"
_ICLR.cc/2026/Conference — ICLR 2026 Poster_

### Official Review · Reviewer_PKB1 · 2025-10-16

**Soundness:** 4
**Presentation:** 3
**Contribution:** 3
**Rating:** 8
**Confidence:** 4

**Summary:**

The paper presents a novel and well-motivated framework that integrates Bayesian surprise into video-language models, enabling belief-driven reasoning and adaptive frame selection. The proposed SPIKE and SPIKE-RL frameworks not only introduce interpretable belief tracking but also deliver consistent improvements across surprise localization and general video understanding benchmarks. The idea of quantifying model surprise to guide frame sampling is elegant and cognitively inspired, bridging the gap between passive perception and proactive understanding in Video-LLMs.

**Strengths:**

1. The idea of connecting Bayesian surprise to video understanding is elegant and intuitive. Whle the proposed method is simple to plug in and does not require retraining large models. As authors report, results are strong and consistent across both surprise-related and general reasoning tasks.

2. The paper is clearly written and easy to follow; the belief visualizations make the concept concrete.

3. The reinforcement learning extension (SPIKE-RL) adds genuine value and improves diversity and accuracy.

**Weaknesses:**

The computational cost of generating multiple belief trajectories isn’t fully discussed.

**Questions:**

Have you consider the other modality included? (e.g.: Audio).

As you mentioned that the Oops is the negetive side, it seems related the anomaly detection? Have you tried the video anomaly understanding as the downstream application?

---

> ### Author Response · Authors · 2025-11-21
> **Author Response to Reviewer PKB1**
>
> Thank you for recognizing that our paper is well-written, and our method is intuitive. We answer your questions below:
>
> **W1: Discussion of computational cost**
> Regarding the time complexity, we sample F frames uniformly from a T-frame video and generate N hypotheses per frame with batched likelihood evaluations. The total cost is O(F · N), linear in the frame budget F. With GPU parallelization, batching the N hypotheses at each step reduces this to O(F). Therefore, though our approach adds hypothesis generation and prior/posterior scoring at multiple timesteps, we parallelize the generation and scoring of hypotheses, and our approach is linear in the number of timesteps taken to score surprise. Our zero-shot baseline results demonstrate that simply viewing a video as a whole is insufficient for precise surprise localization, as the temporal dynamics require more sophisticated processing. The added computational cost yields substantial performance gains (~9 times improvement over zero-shot methods). We have included the complexity analysis in **Sec 2.3** in the revised version attached.
>
>
> **Q1: Audio**:  While SPIKE focuses only on visual evidence, we agree that audio is a rich source of surprise signals, often serving as a precursor (e.g., a crash heard before a visual aftermath) or a contradiction to visual cues.  In SPIKE, the evidence in our Bayesian update could extend to include audio embeddings or transcripts alongside visual frames. We view this multimodal expansion as a promising direction for future work, particularly for domains like movie understanding where dialogue and sound effects play a crucial role in driving the narrative.
>
>
> **Q2: Anomaly Understanding**
> In this work, we model surprise as a function of an observer’s evolving prior beliefs. From this perspective, surprise arises only when new observations fundamentally shift the VLM's belief state, and our goal is to identify the specific events that trigger these significant updates. This differs from anomaly detection, which typically aims to detect statistically unusual events. While we do not evaluate on standard anomaly benchmarks, our method’s success on the Oops! dataset suggests it may extend well to anomalies. However, the reverse is likely not true: standard anomaly detection frameworks are often ill-equipped to handle high-level semantic surprises (such as those in Mr. Bean videos) or 'positive' surprises (as seen creativity, or humor as seen in FunQA)

---

> > ### Comment · Reviewer_PKB1 · 2025-11-27
> > **Reponse to Authors**
> >
> > You have addressed my concerns, and I will maintain my positive score.

---

### Official Review · Reviewer_vBvH · 2025-10-29

**Soundness:** 3
**Presentation:** 3
**Contribution:** 2
**Rating:** 6
**Confidence:** 4

**Summary:**

This paper introduces SPIKE, an inference-time framework that enables Video-LLMs to detect semantically "surprising" events by quantifying Bayesian Surprise. The core mechanism computes surprise as the KL divergence between belief distributions over textual hypotheses before and after observing new visual evidence. The paper further proposes SPIKE-RL, which cleverly uses reinforcement learning (GRPO) with a video captioning reward signal to optimize the hypothesis generation process, by passing the need for direct hypothesis supervision.

**Strengths:**

1.	The use of  Bayesian surprise for frame selection introduces an interpretable probabilistic belief-tracking mechanism distinct from prior query-based or heuristic methods.
2.	Human-readable hypothesis sets enable qualitative analysis of belief shifts.
3.	Gains on five video understanding tasks show that surprise-aware sampling transfers beyond the surprise domain.
4.	The method is explained very clearly, and the figures (especially Fig 2 and Fig 4) provide excellent visualizations of the architecture and qualitative improvements .

**Weaknesses:**

1.	The model's performance seems highly dependent on the quality and diversity of the belief set. The number of hypotheses is fixed at N=3, which seems very low to capture a complex belief space.
2.	The sensitivity to hyperparameters (W) is not explored. It is unclear how performance changes with different $W$ sizes or if the complex historical summary $H_t$ is truly necessary compared to just using the $W_t$ window.
3.	The entire SPIKE-RL optimization relies on a proxy reward signal from an LLM (LLM-Match, using Olmo-7B-hf). The paper does not provide validation for the reliability or potential biases of this specific LLM judge. If the reward model is flawed, the optimization may lead to a policy (hypothesis generator) that is simply good at "gaming" the Olmo-7B judge, rather than generating genuinely better hypotheses.
4.	The RL training procedure appears computationally prohibitive. For each training sample, it requires $M$ full rollouts , each involving $K$ steps of hypothesis generation/scoring, plus a final caption generation, plus a call to the reward model LLM. This high training cost (multi-model, multi-step) may limit the method's scalability, reproducibility, and adoption.

**Questions:**

1.	How reliable is the LLM-Match reward function across diverse linguistic styles?
2.	Does the reinforcement signal propagate stably when rewards are normalized within small rollout groups?
3.	For the Qwen2.5-VL baseline in Table 1, the prompt in Appendix A.3 asks for a binary 0/1 label . How was this binary output converted into a continuous surprise score suitable for calculating the IoU and Acc metrics, which require finding a "peak" or a "window"?

---

> ### Author Response · Authors · 2025-11-21
> **Author Response to Reviewer vBvH - Part I**
>
> Thanks for your positive feedback on our work and thoughtful questions regarding hyperparameter sensitivity.
>
> **W1 : Hypothesis Set Size**
>
> We fixed our hypothesis set size to 3, as it worked well for surprise detection in the videos in our benchmarks, and the beliefs are restricted, as they are the most likely beliefs given the context of the video. To understand the sensitivity to hypothesis set size, we conducted experiments with varying numbers of hypotheses:
>
> **Oops!:**
>
> | Approach | **# Hypotheses (N)** | **Acc@0.25s** | **Acc@1s** |
> | --- | --- | --- | --- |
> | SPIKE | 3 | 60.0 | 67.30 |
> | SPIKE | 5 | 60.36 | 67.73 |
> | SPIKE | 10 | 60.34 | 67.69 |
>
> **Mr. Bean:**
>
> | **Approach** | **# Hypotheses (N)** | **Acc@0.25s** | **Acc@1s** |
> | --- | --- | --- | --- |
> | SPIKE | 3 | 53.20 | 70.20 |
> | SPIKE | 5 | 53.67 | 70.82 |
> | SPIKE | 10 | 53.7 | 70.86 |
>
> Increasing from 3 to 5 hypotheses yields marginal improvements (~0.5-0.6% on Acc@1s), while further increasing to 10 hypotheses shows diminishing returns. This occurs because the probability mass is already captured by the top 5 hypotheses. and adding more hypotheses does not meaningfully change the KL divergence. We chose H=3 as the default for optimal balance between accuracy and efficiency, though in practice, it is possible to use H=5 for slightly better performance.
>
> **W2: Sensitivity to W and Ablation on historical summary**
> Thank you for raising these important points regarding the role of the historical summary and window size W. We conducted an ablation study removing the historical summary component:
>
> **Oops!:**
> | **Approach** | **Acc@0.25s** | **Acc@1s** |
> | --- | --- | --- |
> | SPIKE | 60.0 | 67.37 |
> | SPIKE w/o Historical Summary | 53.7 | 61.29 |
>
> The historical summary is critical for semantic understanding of surprise. To recognize a surprising fall, the model must first understand the narrative context (e.g., A man is riding a bike smoothly). Without the textual summary, the model relies solely on the visual buffer W, which captures motion but lacks the narrative intent that makes divergences semantically meaningful. The summary enables the model to detect violations of established narrative expectations.
>
> We conducted an ablation study on different values for W and obtain the following results.
>
> **Oops!:**
> | **Approach** | **W** | **Acc@0.25s** | **Acc@1s** |
> | --- | --- | --- | --- |
> | SPIKE | 2 | 59.55 | 67.04 |
> | SPIKE | 4 | 60.0 | 67.37 |
> | SPIKE | 6 | 59.97 | 67.33 |
> | SPIKE | 8 | 59.33 | 67.32 |
> |  |  |  |  |
>
> **Mr.Bean:**
>
> | Approach | W | **Acc@0.25s** | **Acc@1s** |
> | --- | --- | --- | --- |
> | SPIKE | 2 | 52.75 | 69.74 |
> | SPIKE | 4 | 53.20| 70.20 |
> | SPIKE | 6 | 53.35 | 70.38 |
> | SPIKE | 8 | 53.07 | 70.11 |
>
> There is a trade-off between Prior Window W and surprise localization accuracy. We find that shorter windows W=2, may miss key visual priors despite the historical summary, while extremely long windows such as W=8 degrade performance by diluting immediate context with older actions.  We found W=4 -- 6 to be optimal for our tasks. However, W is a hyperparameter and may vary based on the nature of surprise in the downstream task.

---

> ### Author Response · Authors · 2025-11-21
> **Author Response to Reviewer vBvH - Part II**
>
> **W3 and Q1: Human Evaluation & LLM-Eval Biases**
>
> Thanks for raising this question about the LLM-Match score. We conducted a human evaluation to validate the LLM-Match Reward. We ask 3 annotators to rate the match between 100 ground-truth and generated captions from Qwen2.5 VL in a continuous score range of 0-1, similar to the LLM.  We found the mean Spearman’s ρ = 0.72 between human and LLM, indicating a **strong** agreement. For reference, human evaluators correlated with each other with average ρ = 0.76. Hence, LLM-Match agrees with human evaluation nearly as much as human subjects do with one another.
>
> **W4: Training Costs and Scaling**
>
> Regarding the time complexity, we sample F frames uniformly from a T-frame video and generate N hypotheses per frame with batched likelihood evaluations. The total cost is O(F · N), linear in the frame budget F. With GPU parallelization, batching the N hypotheses at each step reduces this to O(F). Therefore, though our approach adds hypothesis generation and prior/posterior scoring at multiple timesteps, we parallelize the generation and scoring of hypotheses, and our approach is linear in the number of timesteps taken to score surprise. Our zero-shot baseline results demonstrate that simply viewing a video as a whole is insufficient for precise surprise localization, as the temporal dynamics require more sophisticated processing. The added computational cost yields substantial performance gains (~9 times improvement over zero-shot methods). We have included the complexity analysis in Sec 2.3.
>
> **Q2: RL training vs rollouts**
>
> Thank you for this thoughtful question about our rollout size. Although our rollout group size (m=3) is smaller than typical RL batch sizes, each rollout generates rich gradient signals by scoring multiple hypotheses across numerous video windows. For example, each rollout may generate 8*3 = 24 hypothesis for 8 timesteps. Throughout training, we monitor reward statistics and KL divergence, observing that the KL between the policy and the reference model remained low and bounded (typically 0.02–0.04 across training). The consistent performance improvements across all datasets in Table 1 empirically validate that our normalization strategy effectively propagates learning signals without introducing instability.
>
> **Q3: Qwen2.5 VL Zero-Shot**
> We compute the metrics as follows. For binary scoring, we count a prediction as correct if any predicted surprising frame falls within 0.25 seconds of ground truth (Acc@0.25s), making it an upper bound for the zero-shot model. IoU is computed by extracting continuous segments of predicted positive labels against ground truth windows. For continuous scoring, we identify peak surprise scores for accuracy and define surprising windows as regions exceeding 0.8× the maximum surprise score.

---

### Official Review · Reviewer_FCCu · 2025-10-30

**Soundness:** 3
**Presentation:** 3
**Contribution:** 3
**Rating:** 6
**Confidence:** 2

**Summary:**

The paper proposes **SPIKE**—an inference-time, **Bayesian-surprise** framework for Video-LLMs—and **SPIKE-RL**, a reinforcement-learning refinement of SPIKE’s hypothesis generation. SPIKE tracks **human-interpretable textual beliefs** about upcoming events and defines surprise as the **divergence between prior and posterior belief distributions** when a new frame arrives; this surprise signal then drives **query-agnostic, surprise-weighted frame sampling** that reallocates a fixed frame budget toward informative moments. SPIKE-RL uses **GRPO** with an **LLM-Match** caption reward to improve the diversity and usefulness of belief trajectories. The approach substantially improves **surprise localization** on **Oops!** and **FunQA**, and delivers consistent gains on **five downstream benchmarks** (e.g., BlackSwan, ExFunTube, VideoMME-S, NextQA) by simply replacing uniform sampling in Qwen2.5-VL with surprise-weighted sampling.

**Strengths:**

* Surprise is formalized as **KL/JSD** between posterior and prior over **textual hypotheses**, yielding a transparent score and human-readable beliefs that explain *why* a segment is surprising.
* Surprise-weighted sampling consistently outperforms uniform and classic shot-boundary heuristics across diverse tasks, with additional improvements from SPIKE-RL.
* SPIKE-RL approaches human [Acc@0.25s](mailto:Acc@0.25s) on **Oops!** and improves **FunQA** IoU, showing robust belief tracking beyond zero-shot baselines.
* Drops in as a **sampling layer** for existing Video-LLMs, leaving architectures untouched and keeping complexity linear in frame budget.

**Weaknesses:**

* SPIKE-RL optimizes with an **LLM-based caption reward** (LLM-Match); potential evaluator bias and reward hacking risks are only partially audited.
* Surprise localization relies on **Oops!**, **FunQA**, and a **Mr. Bean** set with **laughter-based silver labels**; broader domains and richer human annotations would strengthen claims.
* Inference adds hypothesis generation and prior/posterior scoring at multiple timesteps

**Questions:**

1. What additional **human studies or inter-rater reliability analyses** would you require to trust **LLM-Match** as a faithful reward for belief optimization (vs. stylistic matching)?
2. Which **non-humor / non-slapstick** domains and **alternative surprise annotations** (beyond laughter or single-event labels) should be added to convincingly establish general surprise tracking?
3. What **latency/throughput breakdowns** (per-step hypothesis generation, scoring, sampling) and **ablation of frame budget vs. quality** would you need to see to judge practicality for streaming or long-video settings?

---

> ### Author Response · Authors · 2025-11-21
> **Author Response to Reviewer FCCu**
>
> Thank you for recognizing that our work provides human-interpretable explanations and outperforms prior approaches. We address your concerns below.
>
> **W1 & Q1: Human studies on LLM match**
>
> Thanks for raising this question about the LLM-Match score. We conducted a human evaluation to validate the LLM-Match Reward. We ask 3 annotators to rate the match between 100 ground-truth and generated captions from Qwen2.5 VL in a continuous score range of 0-1, similar to the LLM.  We found the mean Spearman’s ρ = 0.72 between human and LLM, indicating a **strong** agreement. For reference, human evaluators correlated with each other with average ρ = 0.76. Hence, LLM-Match agrees with human evaluation nearly as much as human subjects do with one another.
>
> **W2 and Q2: Surprise Localization Domains/Data**
>
> We appreciate this observation about annotating surprising events and expanding beyond humor. Our surprise localization evaluation uses gold-standard labels for Oops! and FunQA and silver-standard labels for Mr. Bean dataset. We acknowledge that expanding evaluation to include more gold-standard annotations across diverse scenarios would strengthen our findings. Promising domains for future data collection include: Video Summarization (e.g., of TV shows), sports highlights and unexpected game moments, egocentric/first-person videos, dashcam footage and AI-generated video content.
> In order to evaluate whether SPIKE-RL works on longer and wider scope of videos, we will evaluate it on a video summarization task (e.g, TvSum) and include the results in our camera ready version.
>
>
> **W3: Complexity**
>
> Regarding the time complexity, we sample F frames uniformly from a T-frame video and generate N hypotheses per frame with batched likelihood evaluations. The total cost is O(F · N), linear in the frame budget F. With GPU parallelization, batching the N hypotheses at each step reduces this to O(F). Therefore, though our approach adds hypothesis generation and prior/posterior scoring at multiple timesteps, we parallelize the generation and scoring of hypotheses, and our approach is linear in the number of timesteps taken to score surprise. Our zero-shot baseline results demonstrate that simply viewing a video as a whole is insufficient for precise surprise localization, as the temporal dynamics require more sophisticated processing. The added computational cost yields substantial performance gains (~9 times improvement), as shown in Table1. We have included the complexity analysis in Sec 2.3 in our revised version.
>
> **Q3: Latency/throughput for practicality**
>
> Our design choices ensure our approach is practical for real-world applications:
>
> - All hypothesis generation and scoring operations run in parallel.
> - Our compute time scales linearly with video length (number of timesteps), making performance predictable.
> - Our summarization-based history maintains constant memory usage rather than accumulating past observations, enabling processing of arbitrarily long videos without memory constraints.
>
> Here is the trade-off of B vs. Surprise localization accuracy (quality) on Oops!.
>
> | **Approach** | **B** | **Acc@0.25s** | **Acc@1s** |
> | --- | --- | --- | --- |
> | SPIKE | 8 | 55.60 | 63.76 |
> | SPIKE | 16 | 58.44 | 66.41 |
> | SPIKE | 32 | 58.91 | 66.83 |
> | SPIKE | 64 | 60.00| 67.37 |
> | SPIKE | 128 | 60.25 | 67.65 |
>
> At 32 frames (half our default budget), performance drops  by ~1.1% drop in Acc@0.25s compared to 64 frames. At extremely low  budgets (F=8), SPIKE achieves 55.6%, resulting in a ~4% drop. Performance saturates beyond 64 frames, with 128 frames yielding only marginal gains (+0.25% Acc@0.25s),  suggesting that we are approaching the performance ceiling on this benchmark.

---

### Official Review · Reviewer_2MSg · 2025-10-31

**Soundness:** 3
**Presentation:** 3
**Contribution:** 3
**Rating:** 6
**Confidence:** 4

**Summary:**

This paper introduces SPIKE-RL, a frameworks that enables Video-LLMs to identify surprising moments in videos through Bayesian belief tracking. The core approach maintains explicit probability distributions over textual hypotheses about what might happen next in a video, computing surprise as the KL divergence between prior and posterior beliefs when new frames are observed. SPIKE-RL further refines this through reinforcement learning (GRPO) that optimizes hypothesis generation based on video captioning quality. The authors demonstrate that their surprise scores correlate strongly with human judgments on surprise localization benchmarks and show that replacing uniform frame sampling with surprise-weighted sampling—which allocates more frames to high-surprise moments—leads to consistent improvements across five downstream video understanding tasks including question answering and temporal reasoning. The method is query-agnostic, computationally efficient, and provides interpretable results through human-readable belief hypotheses, offering a principled alternative to uniform sampling that better mimics human attention to memorable, expectation-violating events.

**Strengths:**

1. Novel and principled approach: Introduces a theoretically grounded Bayesian framework for surprise detection in videos, formalizing belief tracking through explicit probability distributions over human-interpretable textual hypotheses rather than relying on heuristics or purely visual features.
2. The proposed method achieves near-human performance on surprise localization (62.9% vs 62.1% human accuracy on Oops! Acc@0.25s) and significantly outperforms baselines across multiple benchmarks, with 10-fold improvement over zero-shot Qwen2.5-VL.
3. Unlike query-conditioned methods, SPIKE works proactively without requiring downstream questions, and provides human-readable belief hypotheses that make the surprise scores interpretable and debuggable.

**Weaknesses:**

1. The training set is small. SPIKE-RL is trained on only 2,000 videos, which raises concerns about overfitting and generalization. No ablation studies on training set size or analysis of what the model actually learns during RL training.
2. The zero-shot Qwen2.5-VL baseline appears poorly designed (simple binary scoring prompt). Missing comparisons with recent video understanding methods that use attention mechanisms or saliency detection. Shot boundary detection baselines (RGB Histogram, ECR) seem strawmen as they're not designed for semantic surprise.
3. The scope of the method is limited. Only tested on short videos were used and evaluation focuses heavily on unintentional failures (Oops!) which may not generalize to other surprise types
4. Limited discussion of attention mechanisms in video transformers, video saliency detection, or cognitive science literature on prediction error beyond the single Bayesian ToM citation.
5. On VideoMME-S and NextQA, gains are only +2.7% and +1.7%, suggesting the approach primarily benefits surprise-heavy scenarios, limiting broader applicability claims.

**Questions:**

Why is KL divergence the right measure of surprise for this task? JSD mentioned in appendix but not compared

---

> ### Author Response · Authors · 2025-11-21
> **Author Response to Reviewer 2MSg - Part I**
>
> Thank you for your thoughtful feedback, and for recognizing the novelty and theoretical grounding of our work. We address your concerns below.
>
> **W1: Training Set Size and RL training analysis:**
>
> While SPIKE-RL is trained on ~2,000 videos from ActivityNet and Oops, we observe strong generalization across multiple **out-of-distribution (OOD)** benchmarks:
>
> - **Surprise Localization Performance (Table 1):** On both FunQA and Mr. Bean datasets (OOD), SPIKE-RL improves temporal IoU over prior baselines and achieves a 4% gain over the base SPIKE model.
> - **Downstream Task Performance (Table 2):** SPIKE-RL demonstrates consistent improvements ranging from 2-7% across FunQA Captions, ExFunTube Captions, Video-MME, and NextQA, indicating strong transfer beyond training domains.
>
> These gains suggest that with our current training set, the model learns generalizable belief-tracking behaviors rather than memorizing specific training videos. Beyond performance, we analyze how RL training changes the model’s beliefs and include it in Sec. 4.3:
>
> - **Diversity**: SPIKE-RL (40.3%) achieves **7%** higher diversity of beliefs than SPIKE’s (33.5%), demonstrating that RL training enriches the model's belief space with more diverse beliefs.
> - **Human alignment**: Human evaluation shows improved Spearman correlation for SPIKE-RL compared to SPIKE (0.87 vs 0.84), demonstrating better alignment with human judgments of prior and posterior beliefs.
> - **Surprise scoring**: Performance gains on diverse tasks (~3-4% on localization, ~1-2% on downstream tasks) validate that the RL training is beneficial.
>
> Our primary contribution is that we model surprise detection as bayesian belief tracking in VLMs, and demonstrate that RL can further optimize beliefs. The consistent OOD gains and improved belief-quality metrics indicate our approach generalizes effectively at the current scale of training. We acknowledge that scaling training data would likely yield further improvements and aim to explore this in future work.
>
> **W2: Zero-Shot Baselines:**
>
> Thank you for this thoughtful question about the zero-shot baselines. We ran the zero-shot Qwen2.5 VL baseline with multiple prompting strategies to establish additional zero-shot baselines. We test with the model on the following prompts: (i) Binary Scoring: Where the zero-shot model provides a binary label for whether each frame is surprising (ii) Binary Scoring+CoT: The model provides a Chain-of-Thought followed by the binary label per frame (iii) Continuous Scoring: Where the zero-shot model provides a score between 0 to 1 per frame (iv) Continuous Scoring+ CoT, where the model provides a Chain-of-thought, followed by a continuous score per frame (iv) Localize surprising frames : Where the model lists the contiguous range of frames that are surprising, serving as an alternative to binary labels.  We compute the metrics as follows. For binary scoring, we count a prediction as correct if any predicted surprising frame falls within 0.25 seconds of ground truth (Acc@0.25s), making it an upper bound for the zero-shot model. For continuous scoring, we identify peak surprise scores for accuracy.
>
> The results of the zero-shot baselines are as follows:
>
> | Approach | Prompt Strategy | Acc@0.25s | Acc@1s |
> | --- | --- | --- | --- |
> | Qwen2.5VL 0-shot | Binary Scoring  | 6.59 | 9.60 |
> | Qwen2.5VL 0-shot | Binary Scoring + CoT | 6.20 | 9.25 |
> | Qwen2.5VL 0-shot | Continuous scoring | 4.30 | 7.60 |
> | Qwen2.5VL 0-shot | Continuous scoring + CoT | 5.19 | 8.89 |
> | Qwen2.5VL 0-shot | Localize surprising frames | 2.57 | 5.67 |
> | SPIKE | Window-Based Belief tracking | 60.0 | 67.37 |
>
> Despite exploring diverse prompting strategies, SPIKE outperforms all zero-shot approaches by ~9 times. This substantial gap demonstrates that zero-shot prompting methods fail to capture the temporal dynamics essential for surprise detection. We attribute this to the fact that standard VLM instruction tuning emphasizes global video understanding over dense temporal grounding. Furthermore, zero-shot inference lacks the explicit frame-by-frame belief tracking essential for modeling how expectations shift over time.

---

> ### Author Response · Authors · 2025-11-21
> **Author Response to Reviewer 2MSg - Part II**
>
> **W3: Scope of Surprise**
>
> - **Surprise types beyond Oops!:** We evaluate our method on three diverse surprise-related datasets - Oops, FunQA, and Mr.Bean, as detailed in Table 1 of our paper. While Oops! covers negative surprises such as falls, FunQA has positive surprises such as creativity and humor, and Mr. Bean has humorous surprises that require a deeper understanding of the storyline. This indicates that our method generalizes to other surprise types beyond unintentional failures (Oops!).
> - **Scope beyond Surprise domain:** We chose Oops!, FunQA and Mr.Bean due to the availability of gold or silver standard annotations for surprise in these datasets. In order to evaluate whether SPIKE-RL works on longer and wider scope of videos, we will evaluate it on a video summarization task (e.g, TvSum) and include the results in our camera ready version.
>
> **W4: Discussion on Video Saliency and Prediction Error:**  Thank you for pointing this out. We have expanded our related work to discuss video saliency detection, and cognitive science literature on prediction error in the new attached version.
>
> **W5: QA tasks**
> Regarding the comparatively modest improvements on Video-MME and NextQA relative to other benchmarks (e.g., ExFunTube, FunQA), we note that this difference is consistent with the fundamental characteristics of these datasets. Unlike tasks that emphasize salient or unexpected content, Video-MME and NextQA contain questions about routine activities and mundane video frames that do not necessarily benefit from heightened attention to surprising moments. Nevertheless, the fact that our approach still yields measurable improvements on these benchmarks demonstrates its generalizability beyond surprise-centric tasks, even when surprise identification is not the primary requirement.
>
> **Q1: KL vs JS divergence:**
>
> Our approach focuses on the relative ordering of surprise scores to identify peak surprising windows, rather than absolute magnitudes. Since JS divergence is derived from KL divergence and exhibits similar monotonic properties (detailed in the Appendix), the choice between these metrics does not affect surprise localization performance. Both preserve the rank ordering necessary for detecting surprising peaks and windows, making them functionally equivalent for our task. In SPIKE, we primarily use KL divergence for the frame sampling and metrics.  JSD is used in our visualization as it is symmetric and bounded (0-1) compared to KL.

---

### Meta-Review · Area_Chair_8kVP · 2026-01-09

**Summary:**

This paper proposes a framework that enables Video-LLMs to identify surprising moments in videos through Bayesian belief tracking.  The paper further proposes to use reinforcement learning with a video captioning reward signal to optimize the hypothesis generation process. The reviewers appreciate the proposed novel and principled approach that yields a transparent score and human-readable beliefs that explain why a segment is surprising.  They also appreciate good outperformance showing robust belief tracking beyond zero-shot baselines.  On the other hand, raised concerns are on validation about overfitting and limited generalization, insufficient zero-shot baselines, ablation of frame budget vs quality, hypothesis set size, etc. Extension to multimodal expansion and applicability to anomaly detection were also raised for arguments.  The authors addressed the raised concerns one by one in their rebuttal, and they also provided additional experimental results to convince the reviewers.  AC thinks that the arguments addressed in the rebuttal are convincing and almost all the concerns were resolved.  Accordingly, the paper should be accepted.

**Reviewer Concerns:**

As written above, insufficiency of validations to fully support the claims were the main concern raised by the reviewers. They were properly addressed by the rebuttal.  Inclusion of audio modality and application to anomaly understanding as the downstream task were provided in terms of arguments only and were left for future work.  However, AC thinks that this does not degrade the contribution of the paper.

**Reviewer Scores:**

All the reviewers would keep their initial scores or raise their scores to 8.

---

### Decision · Program_Chairs · 2026-01-26

Accept (Poster)